# Week 96 Results of Switching from Tenofovir Disoproxil Fumarate-Based Antiretroviral Therapy to Coformulated Elvitegravir, Cobicistat, Emtricitabine, and Tenofovir Alafenamide among HIV/Hepatitis B Virus-Coinfected Patients

Yu-Shan Huang,[a] Chien-Yu Cheng,[b,c] Hsin-Yun Sun,[a] Shu-Hsing Cheng,[b,d] Po-Liang Lu,[e] Chen-Hsiang Lee,[f] Yuan-Ti Lee,[g,h] Hung-Chin Tsai,[i] Chia-Jui Yang,[j,k] Chun-Eng Liu,[l] Bo-Huang Liou,[m] Shih-Ping Lin,[n] Sung-Hsi Huang,[o,p] Mao-Wang Ho,[q] Hung-Jen Tang,[r,s] Chien-Ching Hung,[a,p,t] on behalf of the Taiwan HIV Study Group

[a]Department of Internal Medicine, National Taiwan University Hospital and National Taiwan University College of Medicine, Taipei, Taiwan

[b]Department of Infectious Diseases, Taoyuan General Hospital, Ministry of Health and Welfare, Taoyuan, Taiwan

[c]School of Public Health, National Yang Ming Chiao Tung University, Taipei, Taiwan

[d]School of Public Health, Taipei Medical University, Taipei, Taiwan

[e]Department of Internal Medicine, Kaohsiung Medical University and College of Medicine, Kaohsiung, Taiwan

[f]Department of Internal Medicine, Kaohsiung Chang Gung Memorial Hospital and Chang Gung University College of Medicine, Kaohsiung, Taiwan

[g]Department of Internal Medicine, Chung Shan Medical University Hospital, Taichung, Taiwan

[h]School of Medicine, Chung Shan Medical University, Taichung, Taiwan

[i]Department of Internal Medicine, Kaohsiung Veterans General Hospital, Kaohsiung, Taiwan

[j]School of Medicine, National Yang Ming Chiao Tung University, Taipei, Taiwan

[k]Department of Internal Medicine, Far Eastern Memorial Hospital, New Taipei City, Taiwan

[l]Department of Internal Medicine, Changhua Christian Hospital, Changhua, Taiwan

[m]Department of Internal Medicine, Hsinchu MacKay Memorial Hospital, Hsinchu City, Taiwan

[n]Department of Internal Medicine, Taichung Veterans General Hospital, Taichung, Taiwan

[o]Department of Internal Medicine, National Taiwan University Hospital Hsinchu Branch, Hsinchu, Taiwan

[p]Department of Tropical Medicine and Parasitology, National Taiwan University College of Medicine, Taipei, Taiwan

[q]Department of Internal Medicine, China Medical University Hospital, Taichung, Taiwan

[r]Department of Internal Medicine, Chi Mei Medical Center, Tainan, Taiwan

[s]Department of Health and Nutrition, Chia Nan University of Pharmacy and Sciences, Tainan, Taiwan

[t]Department of Internal Medicine, National Taiwan University Hospital Yunlin Branch, Yunlin, Taiwan

**ABSTRACT** Data regarding the durability of tenofovir alafenamide (TAF)-containing antiretroviral therapy (ART) in maintaining hepatitis B virus (HBV) viral suppression among HIV/HBV-coinfected patients are limited. Between February and October 2018, 274 HIV/HBV-coinfected participants who had achieved HIV RNA of <50 copies/mL with tenofovir disoproxil fumarate (TDF)-containing ART and switched to elvitegravir/cobicistat/emtricitabine/TAF were prospectively enrolled. Serial plasma HIV and HBV viral loads, HBV and hepatitis D virus (HDV) serology, renal parameters, metabolic profiles, and bone mineral density (BMD) were assessed through 96 weeks. At baseline and weeks 48, 72, and 96, 5.8%, 5.1%, 5.8%, and 5.1% of the participants had plasma HBV DNA of ≥20 IU/mL, and 0%, 0.7%, 1.5%, and 2.2% had HIV RNA of ≥50 copies/mL, respectively. Hepatitis B surface antigen (HBsAg) loss occurred in 1.5% of 274 participants, and hepatitis B e-antigen (HBeAg) loss or seroconversion occurred in 14.3% of 35 HBeAg-positive participants. Compared with baseline, the median urine protein-to-creatinine ratio (79 versus 63 mg/g, $P < 0.001$) and $\beta$2-microglobulin-to-creatinine ratio (165 versus 83 $\mu$g/g, $P < 0.001$) continued to decrease at week 96. BMD of the spine and hip slightly increased (mean change, +0.9% and +0.5%, respectively). The median triglycerides, total cholesterol, low-density lipoprotein (LDL)-cholesterol and high-density lipoprotein (HDL)-cholesterol increased from baseline to week 96 (116 versus

Address correspondence to Chien-Ching Hung, hcc0401@ntu.edu.tw, or Hung-Jen Tang, 8409d1@gmail.com.

The authors declare a conflict of interest. Chien-Ching Hung has received research support from Gilead Sciences, Merck, and ViiV and speaker honoraria from Gilead Sciences and ViiV, and served on advisory boards for Gilead Sciences and ViiV. Hsin-Yun Sun has received research support from Gilead Sciences. Other authors report no conflicts of interest.

141, 166 versus 190, 99 versus 117, and 42 versus 47 mg/dL, respectively; all $P < 0.001$), and most of the increases occurred in the first 48 weeks of the switch. Our study showed that switching from TDF-containing ART to elvitegravir/cobicistat/emtricitabine/TAF maintained HBV and HIV viral suppression through 96 weeks among HIV/HBV-coinfected patients. Proteinuria continued to improve, while fasting lipids increased and BMD stabilized at 96 weeks after the switch.

**IMPORTANCE** Elvitegravir/cobicistat/emtricitabine/tenofovir alafenamide as a maintenance therapy showed durable and high rates of viral suppression for HIV/HBV-coinfected patients, with only 5.1% and 2.2% of patients having HBV DNA of ≥20 IU/mL and HIV RNA of ≥50 copies/mL, respectively, at 96 weeks. Our study fills the data gap on the long-term clinical effectiveness of tenofovir alafenamide-containing antiretroviral therapy in people living with HIV who have HBV coinfection.

**KEYWORDS** viral hepatitis, antiretroviral therapy, bone mineral density, hepatitis D virus, hyperlipidemia, proximal renal tubulopathy, tenofovir

Hepatitis B virus (HBV) infection is highly endemic in Southeast Asia, and about 10% of people living with HIV (PLWH) in Taiwan are affected by chronic hepatitis B (1). PLWH who have HBV coinfection are at higher risk of cirrhosis of the liver and death (2, 3), while achieving sustained HBV suppression by long-term antiviral therapy is shown to reduce liver-related complications (4). The ultimate goal of HBV therapy is seroclearance of hepatitis B surface antigen (HBsAg), but it occurs infrequently in PLWH (5). Therefore, lifelong antiretroviral therapy (ART) with regimens containing nucleos(t)ide reverse transcriptase inhibitors (NRTIs) that show potent activity against HBV such as tenofovir disoproxil fumarate (TDF) or tenofovir alafenamide (TAF) are recommended for PLWH with HBV coinfection (6).

TAF and TDF are both prodrugs of tenofovir that can inhibit HIV and HBV replication. Compared to TDF, TAF results in a 90% reduction of tenofovir plasma concentration and has been shown to possess more favorable effects on kidney and bone markers (7). Switching from TDF to TAF leads to increases of bone mineral density (BMD) and decreases of proteinuria but elevations of fasting lipids in both PLWH and HBV-monoinfected populations (8–10). While only a limited number of studies were conducted with TAF-containing regimens among PLWH with HBV coinfection, similar changes were reported (11, 12). The increases in lipid parameters have raised the concern about risks of cardiovascular events or metabolic syndromes (13, 14). Currently, the follow-up durations of most studies were relatively short. The persistence of improvement in renal and bone parameters and the clinical impact of elevated fasting lipids remain to be investigated.

In HBV-monoinfected patients, treatment with TAF showed noninferior efficacy in achieving viral suppression, similar mean declines in quantitative HBsAg and higher rates of alanine aminotransferase (ALT) normalization at week 48 compared to those who continued TDF (10, 15). HBsAg loss or hepatitis B e-antigen (HBeAg) seroconversion occurred in less than 10% of the participants after the switch to TAF (10). However, the evaluation of serologic end points of HBV to TAF-containing ART in PLWH with HBV coinfection were often incomplete, especially for the quantitative HBsAg, and long-term observation has been lacking (11, 16, 17).

In this multicenter prospective study, we reported the 96-week results of HIV and HBV viral suppression and evolution of proteinuria, fasting lipids, and BMD after switching from TDF-containing ART to coformulated elvitegravir, cobicistat, emtricitabine, and TAF (E/c/F/TAF) in PLWH coinfected with HBV.

## RESULTS

**Participants.** A total of 274 HIV/HBV-coinfected participants were included. The clinical characteristics of the participants and the study outcomes at week 48 have been previously presented (12). In summary, HIV-suppressed PLWH with HBV coinfection who switched from TDF-containing ART to E/c/F/TAF achieved high rates of HBV

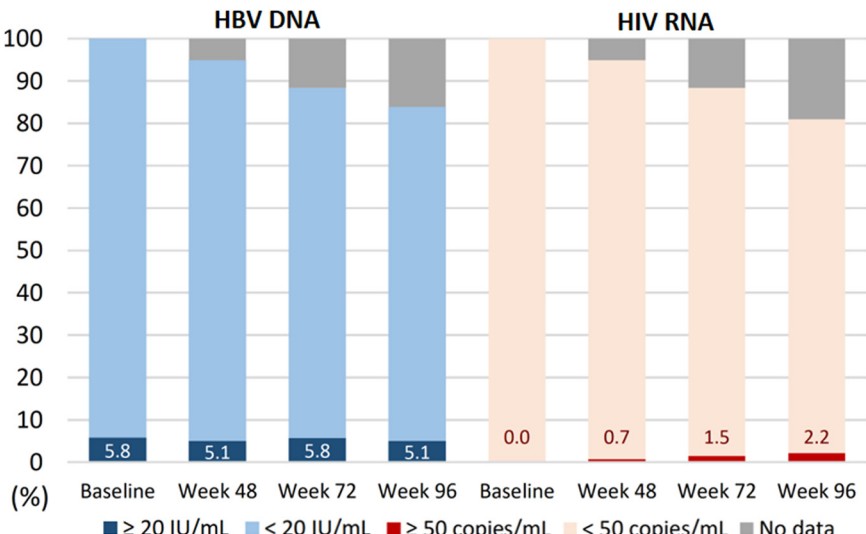

**FIG 1** Proportions of plasma HBV DNA of ≥20 IU/mL and HIV RNA of ≥50 copies/mL in the participants after switching to coformulated elvitegravir, cobicistat, emtricitabine, and tenofovir alafenamide from baseline to week 96 in FDA snapshot analysis.

DNA of <20 IU/mL (89.8%) and HIV RNA of <50 copies/mL (94.2%) at week 48. Also, the participants showed improvement of proteinuria and BMD of the spine and hip but increased fasting lipid levels at weeks 24 and 48.

In this extension study, 230 (83.9%) of 274 participants completed the 96 weeks of follow-up (see Fig. S1 in the supplemental material). A total of 13 (4.7%) and 31 (11.3%) participants dropped out of the study before and after week 48, respectively. After week 48, 31 participants withdrew from the study because of withdrawal of consent (*n* = 12), drug-drug interaction with E/c/F/TAF (*n* = 8), loss to follow-up (*n* = 8), and adverse effects (*n* = 3; hyperlipidemia, occurrence of non-ST-elevation myocardial infarction, and depression). Six participants switched to other TAF-containing ART after 48 weeks and continued in the study.

**Virologic, serologic, and biochemical responses.** The proportions of the participants with HBV DNA of ≥20 IU/mL at weeks 72 and 96 were 5.8% and 5.1%, respectively. At each follow-up time point, the proportion of the participants who had plasma HIV RNA of ≥50 copies/mL ranged from 0.7% to 2.2% (Fig. 1 and Table S1). During the 96-week follow-up period, 7 (2.6%) participants had >1-log$_{10}$ increases in HBV DNA from having achieved <20 IU/mL, and the levels of HBV viral rebound ranged from 209 to 11,400 IU/mL. One participant who had a high HBV DNA load (11,400 IU/mL) at week 72 also showed an increase of plasma HIV RNA from <50 to 1,231,078 copies/mL, suggesting poor treatment adherence. For this participant, E/c/F/TAF was replaced by coformulated bictegravir, emtricitabine, and tenofovir alafenamide (BIC/F/TAF) plus darunavir/cobicistat, and his HBV and HIV loads in subsequent testing were <20 IU/mL and 296 copies/mL, respectively. The other 6 participants continued E/c/F/TAF, and 4 of them reachieved HBV DNA of <20 IU/mL.

The median HBsAg level continued to decrease from 656 IU/mL (interquartile range [IQR], 95 to 1,590) at week 48 to 566 IU/mL (IQR, 68 to 1,495) at week 96 (*P* < 0.001). Participants with positive HBeAg had higher median HBsAg levels at enrollment, and they also showed greater declines in HBsAg levels than HBeAg-negative participants from week 48 to week 96 (Fig. S2). Loss of HBsAg occurred in 4 (1.5%) participants. Among the 35 HBeAg-positive participants, 2 (5.7%) lost HBeAg without positive anti-HBe antibody and 3 (8.6%) had HBeAg seroconversion. For participants who had ALT levels greater than the upper limit of normal at enrollment, 56.4% achieved ALT normalization by central laboratory criteria and 50% by 2018 AASLD criteria (18) at week 96 (Table S1).

**Renal and lipid parameters.** The median estimated glomerular filtration rate (eGFR) decreased from 94.9 mL/min/1.73 m$^2$ at week 48 to 93.2 mL/min/1.73 m$^2$ at week 96

($P$ = 0.002) (Table S2). Participants with an eGFR of ≥90 mL/min/1.73 m², those between 60 and 89 mL/min/1.73 m², and those between 30 and 59 mL/min/1.73 m² at baseline experienced mean changes in eGFR of –7.4%, +0.9%, and +9.4%, respectively. In multivariate logistic regression analysis, use of ART without creatinine transport inhibitors, such as rilpivirine, dolutegravir, bictegravir, cobicistat, or ritonavir, before the switch (adjusted odds ratio [aOR], 3.135; 95% confidence interval [CI], 1.013 to 9.697; $P$ = 0.047) and higher urine protein-to-creatinine ratio (UPCR) (per 1-mg/g increase, aOR 1.005; 95% CI, 1.001 to 1.008; $P$ = 0.007) at baseline were associated with higher risk of an eGFR decline of over 15% at 96 weeks of E/c/F/TAF treatment, while higher CD4 counts at baseline conferred a lower risk (per-1 cell/$\mu$L increase, aOR 0.997; 95% CI, 0.995 to 1.000; $P$ = 0.019) (Table S3).

The UPCR and urine $\beta$2-microglobulin-to-creatinine ratio continued to decline from week 48 to week 96, whereas the decrement of urine albumin-to-creatinine ratio (UACR) was similar between week 48 and week 96 ($-$15.3% versus $-$12.5%) (Fig. 2A). The prevalence of proteinuria by urine dipstick tests decreased from 19.0% at baseline to 3.9% at week 96. Similarly, the proportion of participants with UPCR of ≥150 mg/g decreased from 12.1% at baseline to 7.9% at week 96 (Fig. 2B). The participants with abnormal UPCR (≥150 mg/g) at baseline experienced greater declines in UPCR at week 96 compared to those with normal UPCR at baseline (median percentage change of UPCR: $-$52.7% versus $-$19.5%, $P$ < 0.001).

The median levels of triglycerides, total cholesterol, low-density lipoprotein (LDL)-cholesterol, and high-density lipoprotein (HDL)-cholesterol remained steady from week 48 to week 96. The median changes of triglycerides, total cholesterol, LDL-cholesterol, and HDL-cholesterol from baseline to week 96 were 26.5, 24.5, 17.0, and 5.0 mg/dL, respectively, and the median percentage increases were 24%, 15%, 18%, and 12%, respectively (Fig. 3). No statistically significant changes were found in fasting blood glucose and glycated hemoglobin (HbA1C) levels at week 96 (Table S2). Of the 90 participants with follow-up weight measurements, the median weight was the same at week 48 (72 kg; IQR, 64 to 79) and week 96 (72 kg; IQR, 65 to 80). The median percentage increase in weight was 4.1% at week 96.

**BMD assessment.** BMD assessment was performed in 138 participants at week 96. Among these participants, the mean BMD of the spine and hip increased and peaked at week 48 (+2.0% and +1.5%, respectively) and slightly declined but were still higher at week 96 than those at baseline (+0.9% and +0.5%, respectively) (Fig. 4A and Table S4). For participants who had paired BMD assessments at baseline and week 96, we inspected the changes of T-score categories after switching to TAF-containing ART. At week 96, 44.8% of the participants who had osteopenia of the spine before the switch regressed to a normal T-score, and 33.3% of those who had osteoporosis of the spine regressed to osteopenia. A similar proportion of changes was observed for T-scores of the hip (Fig. 4B).

**Prevalence of hepatitis D virus (HDV) coinfection.** At baseline, 40 (14.6%) participants tested positive for anti-HDV IgG. Injecting drug use (32.5% versus 3.0%, $P$ < 0.001), positive anti-HCV (41.0% or 8.8%, $P$ < 0.001), and elevated serum rapid plasma reagin (RPR) titers at baseline were more commonly found among anti-HDV-positive than anti-HDV-negative participants (57.5% versus 41.6%, $P$ = 0.061) (Table S5). For participants who continued to participate in the follow-up through week 96, HDV seroprevalances at weeks 48, 72, and 96 were 14.2%, 12.4%, and 13.4%, respectively. The proportions of the participants with HDV viremia ranged from 2.3% to 5.4% at each follow-up time point, and their mean HDV RNA levels at baseline and weeks 48 and 96 were 8.6, 8.8, and 7.5 $\log_{10}$ copies/mL, respectively (Fig. 5). During the study period, anti-HDV seroconversion occurred in 5 (2.1%) of 243 anti-HDV-negative participants. Of the 5 participants with HDV seroconversion, 3 had HDV viremia before seroconversion, 2 had ALT flares to 2-fold or greater of the upper limit of normal, and 2 had a ≥4-fold increase of RPR titers. The incidence rate of HDV infection was estimated to be 11.5 per 1,000 person-years of follow-up.

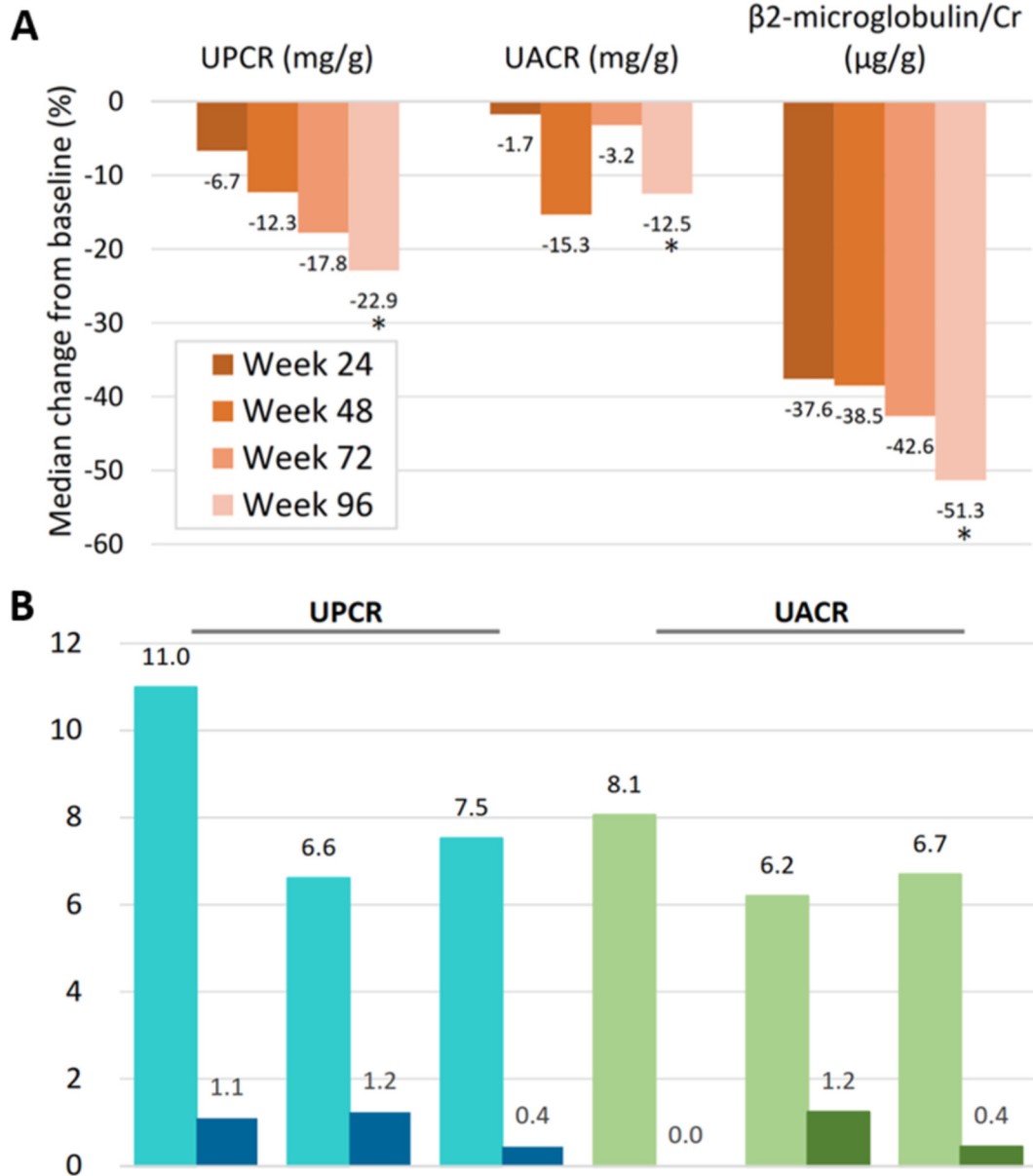

**FIG 2** (A and B) Changes in (A) median quantitative proteinuria and (B) proportion of the participants with abnormal quantitative proteinuria from baseline to week 96. The asterisk (*) indicates that the median level at week 96 is statistically significantly different from that at baseline ($P < 0.05$). UPCR, urine protein-to-creatinine ratio; UACR, urine albumin-to-creatinine ratio; Cr, creatinine.

## DISCUSSION

In this prospective cohort study of PLWH with HBV coinfection, the switch of TDF-containing ART to E/c/F/TAF could maintain high rates of HIV and HBV viral suppression, with only 2% and 5% of participants having plasma HIV RNA of ≥50 copies/mL and HBV DNA of ≥20 IU/mL, respectively, throughout 96 weeks. Improved proteinuria was observed, and the decrease in UPCR was greater among the participants who had UPCR of ≥150 mg/g at baseline. Participants also had increased BMD and fasting lipids, which stabilized after the first 48 weeks of TAF-containing ART.

TAF as a sequential therapy showed a durable antiviral effect in HBV-monoinfected

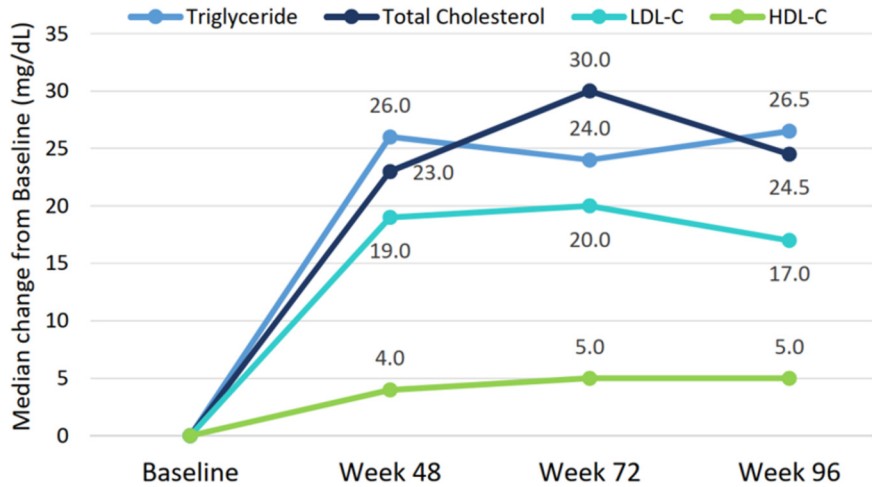

**FIG 3** Changes in lipid profile from baseline through week 96. LDL-C, low-density lipoprotein cholesterol; HDL-C, high-density lipoprotein cholesterol.

patients; over 98% of the participants achieved HBV DNA of ≤20 IU/mL at week 96 (19). Our study was the first to demonstrate the durability of HBV suppression by TAF-containing ART (E/c/F/TAF) among PLWH. Viral rebound occurred rarely, and high HBV viral breakthrough was likely to be associated with nonadherence in our study. In a 96-week observational study, no TAF-associated HBV resistance mutations were found in patients with virologic breakthrough (20). TAF-containing ART was also effective to maintain high rates of HBV viral suppression in PLWH with lamivudine-resistant HBV, whereas 94.9% of the patients achieved HBV DNA of <20 IU/mL after a 96-week treat-

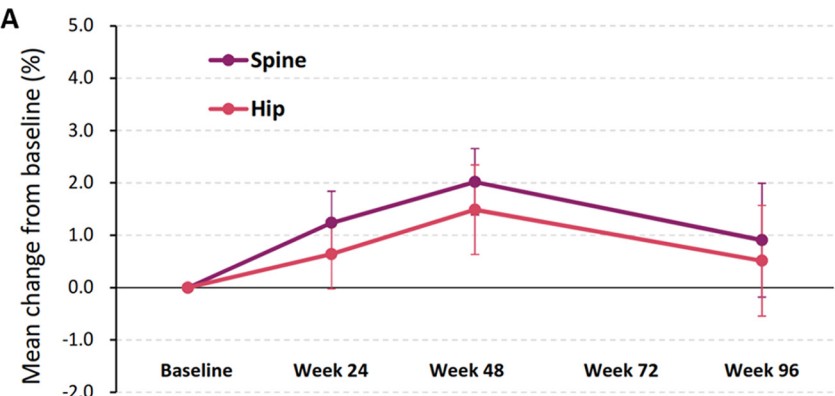

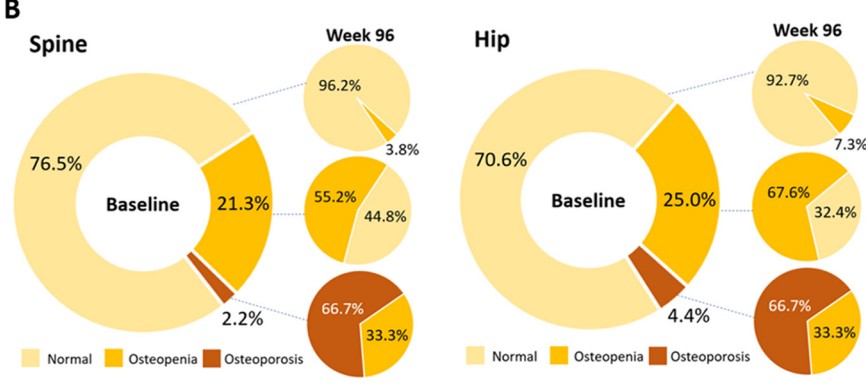

**FIG 4** (A) Changes in bone mineral density of the spine and hip (mean [95% confidence interval]). (B) Changes of T-score categories (normal, osteopenia, and osteoporosis) of the spine and hip from baseline to week 96.

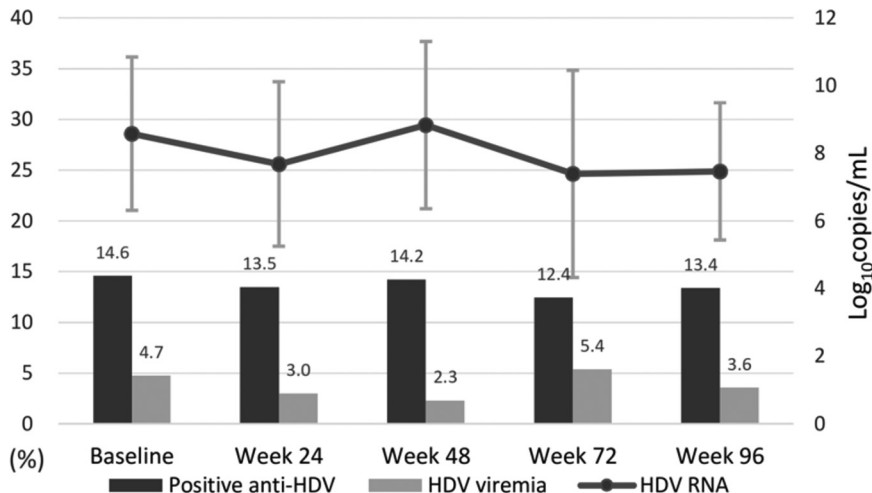

**FIG 5** Proportions of the participants with anti-HDV positivity and HDV viremia, and mean HDV RNA load of the participants testing positive for anti-HDV antibody and HDV RNA.

ment course (21). A longitudinal, single-arm study in HBV-monoinfected patients comparing the rates of HBV DNA of <20 IU/mL 96 weeks before and after switching from TDF to TAF revealed significant improvement in HBV viral suppression after the switch to TAF (22). In a randomized controlled trial, BIC/F/TAF achieved higher rates of HBV DNA of <29 IU/mL (63% versus 43%), HBsAg loss (13% versus 6%), and HBeAg loss (26% versus 14%) than dolutegravir plus TDF/FTC in treatment-naive PLWH with HBV coinfection at week 48, though the trends of HBV DNA declines were similar between the two groups of participants (23). However, the difference was not demonstrated in two randomized trials in treatment-naive HBV-monoinfected patients who initiated TAF or TDF (24, 25). Limited by the different study populations and ages of the participants enrolled in these trials, more studies are warranted to confirm whether TAF has superior anti-HBV activity to TDF among PLWH.

Nephrotoxicity, such as proximal renal tubulopathy, has been the major concern with long-term exposure to TDF (26). Replacement of TDF with non-tenofovir drug or TAF is suggested when proximal renal tubulopathy occurs (27). Among PLWH with HBV coinfection who received TDF-containing ART, our results supported the renal safety of switching from TDF to TAF among PLWH with HBV, especially for those who had moderate or severe proteinuria. In our study, the participants with estimated glomerular filtration rate (eGFR) of <90 mL/min/1.73 m$^2$ at baseline experienced slight increases of eGFR at 96 weeks. Similar findings have been shown among PLWH that switching TDF to TAF resulted in improvement of eGFR among individuals with eGFR of <60 or 60 to 89 mL/min/1.73 m$^2$ (17, 28). It is worth noting that 23.8% of our participants with eGFR of ≥90 mL/min/1.73 m$^2$ at baseline showed declines of eGFR to 60 and 89 mL/min/1.73 m$^2$ (Fig. S3). With a recent report of TAF-related nephrotoxicity (29), the findings of our study suggest that regular monitoring of renal function is needed in PLWH on TAF-containing regimens, particularly in patients with underlying renal abnormalities.

Our study depicting the trends of fasting lipids after switching to TAF-containing ART showed that the increases in lipids stabilized after 48 weeks. However, we did not record the information about lifestyle modifications or prescription of lipid-lowering agents during the 96-week period, both of which might contribute to the findings of reaching a plateau of fasting lipid levels. In our study, two of three adverse events leading to discontinuation of E/c/F/TAF after 48 weeks were associated with hyperlipidemia. In randomized controlled trials, similar proportions of treatment-naive PLWH starting TAF- or TDF-containing ART initiated lipid-lowering medications (19% versus 21%), and there were no differences between the two groups in terms of shifts in categorical risk of atherosclerotic cardiovascular disease (14, 30). However, real-world data on treatment-experienced PLWH revealed

higher rates of prescription of lipid-lowering drugs in PLWH receiving E/c/F/TAF than those receiving E/c/F/TDF (11.9% versus 4.7%, $P = 0.047$) (13). While the conflicting results may be related to different study designs and populations, assessment of lipids and the risk for cardiovascular disease is warranted for timely initiation of lipid-lowering medications, particularly when PLWH who have achieved viral suppression are ageing. The prescription pattern of lipid-lowering drugs among PLWH in real-world settings should also be monitored and evaluated for the adherence to guidelines on primary prevention of cardiovascular disease. Another concern is the impact of increased fasting lipids on the liver. Hyperlipidemia was found to be associated with fatty liver disease in PLWH with HBV coinfection (31). Steatosis alone was independently correlated with higher ALT after adjustment for HBV DNA level (31). Prior studies revealed that the use of TAF or integrase inhibitors but not TDF increased the risk of steatosis in PLWH (32, 33). However, the change in fasting lipids and liver pathology after modifying ART regimens and their long-term clinical impact have not been investigated in PLWH with HBV coinfection.

The HDV seroprevalence was stable through 96, weeks and the incidence rate of HDV seroconvertion in the present study was lower than our earlier estimation between 2007 and 2012 (11.5 versus 13.3 per 1,000 person-years) (34). Nevertheless, high HDV RNA levels were detected in participants with viremia, implying the increased risks of onward HDV transmission. Although the comparisons of RPR-positive rate among anti-HDV-positive and anti-HDV-negative participants in our study did not reach statistical significance (57.5% versus 41.6%), our findings suggested a tendency for HDV to be sexually transmitted among PLWH. Two recent studies proposed that sexual contact has become an important route for HDV transmission (34, 35). The possibility of HDV superinfection should be considered in PLWH with HBV coinfection who recently acquired sexually transmitted diseases. Moreover, high-risk patients may require regular screening for HDV. Nonetheless, prospective studies are needed to clarify the major transmission route of HDV and the association between sexual activities and acquisition of HDV in PLWH with HBV coinfection.

Our study had several limitations. First, there was no comparator group who continued TDF-containing ART. Second, as mentioned above, the numbers and intensities of lipid-lowering agents prescribed during the study period were not included for analysis. However, in published research, about 10% to 20% of PLWH would initiate lipid-lowering drugs while starting or switching to TAF (13, 14, 36). Therefore, the use of lipid-modifying agents might only partially explain the stabilized trends of lipids in our study. Third, PLWH with HBV coinfection who had an eGFR of less than 30 mL/min/1.73 m$^2$ were not included. Therefore, the benefit of improvement in proteinuria with the switch to TAF-containing ART could not be evaluated in patients with severely impaired renal function. Fourth, more participants discontinued the study between week 48 and week 96, which was influenced by lockdown and border control during the early phase of the COVID-19 pandemic and the introduction of BIC/F/TAF in Taiwan. BIC/F/TAF was associated with fewer drug-drug interactions than E/c/F/TAF, and BIC/F/TAF as a maintenance therapy demonstrated low rates of low-level viremia and virologic failure in virally suppressed PLWH (37). Thus, many participants were switched to BIC/F/TAF and withdrew from the subsequent follow-up. Similarly, E/c/F/TAF is seldom used nowadays in many countries. Therefore, the results of HIV suppression and changes in serum creatinine or weight in this study may not be generalizable to PLWH who switch from TDF-containing ART to BIC/F/TAF or dolutegravir plus F/TAF. Finally, the participants who tested HDV seronegative during the 96 weeks of follow-up were not screened for HDV RNA, and the prevalence of HDV viremia could be underestimated.

In conclusion, switching TDF-containing ART to E/c/F/TAF could maintain durable HIV and HBV viral suppression among PLWH with HBV coinfection, and proteinuria continued to improve. Improvement of BMD and the increases in fasting lipids stabilized between week 48 and week 96 of the switch. Long-term follow-up is needed to evaluate the evolutions of lipids and cardiovascular risks.

## MATERIALS AND METHODS

**Study design and participants.** This study enrolled HIV-suppressed PLWH with HBV coinfection who switched from TDF/emtricitabine (FTC)- or TDF plus lamivudine (3TC)-based ART to E/c/F/TAF as a maintenance treatment at 13 hospitals designated for HIV care around Taiwan from February to October 2018. The design, inclusion, and exclusion criteria of the study have been previously described (12). Following the evaluations at baseline and 24 and 48 weeks of E/c/F/TAF, the participants who continued E/c/F/TAF or switched to other TAF-containing ART were monitored for plasma HBV DNA and HIV RNA, HBV serology, anti-HDV IgG, liver and renal functions, urine protein, lipid profiles, fasting glucose, and HbA1C at weeks 72 and 96. BMD assessment was performed only at week 96. The study was approved by the research ethics committee or institutional review board of all participating hospitals. All participants provided written informed consent.

The primary endpoint of this follow-up study was the proportions of participants who maintained undetectable plasma HBV DNA (<20 IU/mL) and plasma HIV RNA (<50 copies/mL) at weeks 72 and 96 after switching from TDF- to TAF-containing ART. Additional endpoints at week 96 included changes in aminotransferases, quantitative HBsAg levels, and rates of HBeAg loss, HBeAg seroconversion, and HDV seroconversion while the participants continued to receive TAF-containing ART. "HBeAg loss" refers to the change from HBeAg positivity and anti-HBe negativity at baseline to HBeAg negativity without the development of anti-HBe after the baseline visit. "HBeAg seroconversion" refers to the change from HBeAg positivity and anti-HBe negativity at baseline to HBeAg negativity and anti-HBe positivity after the baseline visit.

**Laboratory investigations and BMD assessment.** Laboratory tests of HBV serologic markers included HBsAg, anti-HBs, HBeAg, and anti-HBe (Abbott Laboratories, Abbott Park, IL, USA). The eGFR was calculated by the CKD Epidemiology Collaboration (CKD-EPI) equation. The measurement of urine protein included UPCR, UACR, and urine $\beta2$-microglobulin-to-creatinine ratio (AnGene Biotechnology Co., Ltd.). Anti-HDV IgG was determined by competitive enzyme immunoassay (Dia.Pro Diagnostic Bioprobes, Srl, Milan, Italy). For participants with a positive anti-HDV IgG, HDV RNA was determined with the use of in-house real-time PCR assay. In brief, the HDV nucleic acid was first extracted from $500-\mu L$ plasma using the QIAamp UltraSens virus kit (Qiagen, Hilden, Germany), and cDNA was reverse transcribed from $10\ \mu L$ eluted RNA using random hexamer primer. The HDV RNA was then determined using a commercialized HDV quantification primer probe set (TIB LightMix kit) (38, 39) with a detection limit of 10 copies/reaction. The amplification target of HDV was 113 bp on HDV delta antigen.

BMD assessment was performed in participants enrolled at four participating hospitals using dual-energy X-ray absorptiometry (Lunar Prodigy; GE Healthcare, Belgium). Osteoporosis and osteopenia were defined by BMD T-score according to WHO criteria (40).

**Statistical analysis.** The continuous variables were presented as medians and interquartile ranges and compared using the Mann-Whitney U test. The categorical variables were compared using the chi-square test or Fisher's exact test. For paired data, categorical variables were compared using McNemar's test, and continuous variables were compared using the Wilcoxon signed-rank test. Logistic regression analysis was performed to identify determinants of an eGFR decline of over 15% at week 96. Variables identified as known predictors by previous studies, or those with a $P$ value of <0.1 in univariate analysis, were entered into the model with a backward stepwise regression approach. All $P$ values were two-sided, and a $P$ value of <0.05 was considered statistically significant. Statistical analyses were performed using SPSS software version 25.0 (SPSS, Inc., Chicago, IL, USA).

**Ethical approval.** The study was approved by the research ethics committee or institutional review board of the 13 participating hospitals (registration numbers 201710056RINB, 107025-F, TYGH107001, 18MMHIS012e, CMUH107-REC2-081, 107-005-E, CS18052, CF18037B, 171203, 10701-008, 201701662A3, KMUHIRB-SV(II)-20170065, and VGHKS18-CT1-18). All participants gave written informed consent before enrollment.

**Data availability.** Deidentified participant-level data will be available on publication of the study. Requests for data should be sent to the corresponding author by e-mail and, on review of the proposed protocol and signing of a data sharing agreement, the data will be made available.

## SUPPLEMENTAL MATERIAL

Supplemental material is available online only.
**SUPPLEMENTAL FILE 1**, PDF file, 1 MB.

## ACKNOWLEDGMENTS

We thank the participants for participating in this study.

This study was supported in part by Gilead Sciences under grant number IN-US-292-4509.

Chien-Ching Hung has received research support from Gilead Sciences, Merck, and ViiV and speaker honoraria from Gilead Sciences and ViiV and served on advisory boards for Gilead Sciences and ViiV. Hsin-Yun Sun has received research support from Gilead Sciences. The other authors report no conflicts of interest.

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
