## [Reviewer comments · Microbiology Spectrum]

Microbiology Spectrum

Week 96 results of switching from tenofovir disoproxil fumarate-based antiretroviral therapy to coformulated elvitegravir, cobicistat, emtricitabine and tenofovir alafenamide among HIV/HBV-coinfected patients

Yu-Shan Huang, Chien-Yu Cheng, Hsin-Yun Sun, Shu-Hsing Cheng, Po-Liang Lu, Chen-Hsiang Lee, Yuan-Ti Lee, Hung-Chin Tsai, Chia-Jui Yang, Chun-Eng Liu, Bo-Huang Liou, Shih-Ping Lin, Sung-Hsi Huang, Mao-Wang Ho, Hung-Jen Tang, and Chien-Ching Hung

Corresponding Author(s): Chien-Ching Hung, National Taiwan University Hospital, and Hung-Jen Tang, Chi Mei Medical Center

Review Timeline:

Submission Date:	December 15, 2022
Editorial Decision:	December 27, 2022
Revision Received:	February 8, 2023
Accepted:	February 28, 2023

Editor: Yuan Pin Hung

Reviewer(s): Disclosure of reviewer identity is with reference to reviewer comments included in decision letter(s). The following individuals involved in review of your submission have agreed to reveal their identity: Francescopaolo Antonucci (Reviewer #3)

Transaction Report:

DOI: <https://doi.org/10.1128/spectrum.05125-22>

December 27, 2022

Dr. Chien-Ching Hung
National Taiwan University Hospital
Department of Internal Medicine
7, Chung-Shan South Road
Taipei 100
Taiwan

Re: Spectrum05125-22 (Week 96 results of switching from tenofovir disoproxil fumarate-based antiretroviral therapy to coformulated elvitegravir, cobicistat, emtricitabine and tenofovir alafenamide among HIV/HBV-coinfected patients)

Dear Dr. Chien-Ching Hung:

Link Not Available

Sincerely,

Yuan Pin Hung

Journals Department
Reviewer comments:

Reviewer #1 (Comments for the Author):

Huang et al report here the 96wk results of switching from a TDF-based regimen to coformulated E/C/F/TAF in 274 HIV/HBV coinfecting Chinese patients. The authors conclude that switching to E/C/F/TAF can maintain HBV and HIV suppression and lead to improvement in proteinuria and BMD. This observational study is interesting because it brings forward virological data after switching to E/C/F/TAF of a large HIV/HBV coinfecting cohort with a long follow-up.

This work has some problems that I need to point out:

Major points

1. This study analysed data of the continuation to 96wk of the same cohort HIV/HBV whose data they had already analysed and published. Authors use an identical approach and even the same Figures adapted to 96 wk as in the work they previously published (Huang et al. J AIDS 2021; 86: 473-481)

2. E/C/F/TAF is rarely used nowadays in Western countries. Patients on E/C/F/TAF are usually switched to BIC/C/F/TAF (Biktyarvy) avoiding cobicistat.

Minor points

1. Could you report the weight changes after switching to E/C/F/TAF at 96 wk? At 48wk there was a significant weight gain of 3 Kg (JAIDS 2021; 86: 473-481). This is an interesting point to report in the present manuscript.

2. Could you explain the differences between HBeAg loss and HbeAg (negative) seroconversion? They look very similar to me (Results section lines 175-176)

3. RPR titers were more elevated among anti-HDV-positive patients but not significantly (57.5% vs. 41.6%, $p=0.061$ not significant). Therefore there is a tendency but these data cannot confirm here that sexual contact is an important route of HDV transmission (Discussion section, lines 305-308)

Reviewer #2 (Comments for the Author):

1. The drop outs should be discussed better in the beginning. 44 out of 274 did not complete the study but a full account is only given for 19 of them.

2. Discuss long term impact of increased levels of fasting lipids for liver pathology in HBV infected people.

3. Shortcomings are reported but dampen impact of the study (eg no monitoring of lipid lowering drugs)

Typo's

Line 84 : having instead of had

Line 319: were not included

Reviewer #3 (Comments for the Author):

Reviewer Comments to Author:

The authors studied the efficacy and safety of switching from tenofovir disoproxil fumarate-based antiretroviral therapy to coformulated elvitegravir, cobicistat, emtricitabine, and tenofovir alafenamide (E/C/F/TAF) in HIV/hepatitis B virus (HBV)-coinfected Asian patients from Baseline to Week 96.

GENERAL OBSERVATIONS

• Similar study, as also mentioned in Ref 12, has already been conducted and published.

In terms of results, you observed in this new unpublished paper same virologic, serologic and biochemical responses compared to the published study; although, in this new paper, you kept observing patients up to 96 weeks.

Considering these results, this new paper should be re-adapted in a form that includes a brief summary of the previous study (by reporting the observed results at 24 and 48 weeks), and the new results related with follow up at 72 and 96 weeks.

Repeating all the work already done in the published paper (from baseline to 48 hours) is unnecessary and can be confusing (e.g. LINE 144, LINE 341).

In conclusion, it is recommended to edit this manuscript by making a summary of the results obtained so far (baseline and week 48 from the study of REF. 12) and to describe the results observed in week 72 and 96, which are the unpublished results.

MAIN MANUSCRIPT

• Please check the Reference 6 in LINE 113 and make sure it is related with the Ref. 6 of the REFERENCE section;

• LINE 132-133: Reference required; please add one or more references related with the low evaluation of the virologic and serologic parameters in the co-infected patients.

Staff Comments:

Preparing Revision Guidelines

Please return the manuscript within 60 days; if you cannot complete the modification within this time period, please contact me. If you do not wish to modify the manuscript and prefer to submit it to another journal, please notify me of your decision immediately so that the manuscript may be formally withdrawn from consideration by Microbiology Spectrum.

Huang et al report here the 96wk results of switching from a TDF-based regimen to coformulated E/C/F/TAF in 274 HIV/HBV coinfecting Chinese patients. The authors conclude that switching to E/C/F/TAF can maintain HBV and HIV suppression and lead to improvement in proteinuria and BMD. This observational study is interesting because it brings forward virological data after switching to E/C/F/TAF of a large HIV/HBV coinfecting cohort with a long follow-up.

This work has some problems that I need to point out:

Major points

1. This study analysed data of the continuation to 96wk of the same cohort HIV/HBV whose data they had already analysed and published. Authors use an identical approach and even the same figures adapted to 96 wk as in the work they previously published (Huang et al. J AIDS 2021; 86: 473-481).

2. E/C/F/TAF is rarely used nowadays in Western countries. Patients on E/C/F/TAF are usually switched to BIC/C/F/TAF (Biktarvy) avoiding cobicistat.

Minor points

1. Could you report the weight changes after switching to E/C/F/TAF at 96 wk? At 48wk there was a significant weight gain of 3 Kg (JAIDS 2021; 86: 473-481). This is an interesting point to report in the present manuscript.

2. Could you explain the differences between HBeAg loss and HBeAg (negative) seroconversion? They look very similar to me (Results section lines 175-176).

3. RPR titers were more elevated among anti-HDV-positive patients but not significantly (57.5% vs. 41.6%, $p=0.061$ not significant). Therefore there is a tendency but these data cannot confirm here that sexual contact is an important route of HDV transmission (Discussion section, lines 305-308).

Reviewer Comments to Author:

The authors studied the efficacy and safety of switching from tenofovir disoproxil fumarate-based antiretroviral therapy to coformulated elvitegravir, cobicistat, emtricitabine, and tenofovir alafenamide (E/C/F/TAF) in HIV/hepatitis B virus (HBV)-coinfected Asian patients from Baseline to Week 96.

GENERAL OBSERVATIONS

- Similar study, as also mentioned in Ref 12, has already been conducted and published.

In terms of results, you observed in this new unpublished paper same virologic, serologic and biochemical responses compared to the published study; although, in this new paper, you kept observing patients up to 96 weeks.

Considering these results, this new paper should be re-adapted in a form that includes a brief summary of the previous study (by reporting the observed results at 24 and 48 weeks), and the new results related with follow up at 72 and 96 weeks.

Repeating all the work already done in the published paper (from baseline to 48 hours) is unnecessary and can be confusing (e.g. LINE 144, LINE 341).

In conclusion, it is recommended to edit this manuscript by making a summary of the results obtained so far (baseline and week 48 from the study of REF. 12) and to describe the results observed in week 72 and 96, which are the unpublished results.

MAIN MANUSCRIPT

- Please check the Reference 6 in LINE 113 and make sure it is related with the Ref. 6 of the REFERENCE section;
- LINE 132-133: Reference required; please add one or more references related with the low evaluation of the virologic and serologic parameters in the co-infected patients.

Feb 7th, 2023

Manuscript Number: Spectrum05125-22R1

Title: **Week 96 results of switching from tenofovir disoproxil fumarate-based antiretroviral therapy to coformulated elvitegravir, cobicistat, emtricitabine and tenofovir alafenamide among HIV/HBV-coinfected patients**

Dear Editor,

Thank you for your kind consideration of our manuscript. We have carefully read and responded point-by-point to the queries and comments from the reviewers. The response to each of the reviewer's comments is incorporated into the revised manuscript, which is underlined. Figures are added or revised per the reviewer's suggestions. We appreciate their thoughtful and helpful review that has helped strengthen the manuscript. Enclosed are point-by-point responses addressing each of the comments and the revised manuscript. We hope that the manuscript is now acceptable for publication. Please do not hesitate to contact me if there are any questions.

Sincerely,

Chien-Ching Hung

Department of Internal Medicine, National Taiwan University Hospital, 7 Chung-Shan

South Road, Taipei, Taiwan, 100

E-mail: hcc0401@ntu.edu.tw

Telephone: +886-2-23123456

Fax: +886-2-23822172

Responses to the Reviewers

Reviewers' Comments

Comments of reviewer #1:

Reviewer #1: Huang et al report here the 96wk results of switching from a TDF-based regimen to coformulated E/C/F/TAF in 274 HIV/HBV coinfecting Chinese patients. The authors conclude that switching to E/C/F/TAF can maintain HBV and HIV suppression and lead to improvement in proteinuria and BMD. This observational study is interesting because it brings forward virological data after switching to E/C/F/TAF of a large HIV/HBV coinfecting cohort with a long follow-up.

This work has some problems that I need to point out:

Comment 1. This study analysed data of the continuation to 96wk of the same cohort HIV/HBV whose data they had already analysed and published. Authors use an identical approach and even the same Figures adapted to 96 wk as in the work they previously published (Huang et. J AIDS 2021; 86: 473-481)

Reply: Thank you for the comment. Studies of longer follow-up durations of HBV to TAF-based antiretroviral regimens are limited, which prompted us to continue the observation by following the same protocol for the 48-week cohort study. Combining the suggestions from Reviewer 2, we revise Results section as follows: The findings observed from baseline to week 48 were briefly summarized (lines 140-146, underlined), and the descriptions in each paragraph focus on data between weeks 48 and 96. In addition, the quantitative proteinuria and bone mineral density (BMD) are analyzed with different approaches to demonstrate the evolution in categories of UPCR, UACR, and T-score at weeks 72 or 96. The revised sentences are listed below (underlined). Figures 1 to 4 are also added or revised, as follows:

- Lines 140-146, summary of previously published study (underlined): “A total of 274 HIV/HBV-coinfected participants were included. The clinical characteristics of the participants and the study outcomes at week 48 have been previously presented (12). In summary, HIV-suppressed PLWH with HBV coinfection who switched from TDF-containing ART to E/c/F/TAF achieved high rates of HBV DNA <20 IU/mL (89.8%) and HIV RNA <50 copies/mL (94.2%) at week 48. Also, the participants showed improvement of proteinuria and BMD of the spine and hip but increased fasting lipid levels at weeks 24 and 48.”
- Lines 155-157 (underlined): “The proportions of the participants with HBV DNA ≥ 20 IU/mL at weeks 72 and 96 were 5.8% and 5.1%, respectively.”
- Lines 161-163 (underlined): “One participant who had high HBV DNA load (11400 IU/mL) at week 72 also showed an increase of plasma HIV RNA from <50 to 1231078 copies/mL, suggesting poor treatment adherence”
- Lines 168-169 (underlined): “The median HBsAg level continued to decrease from 656 IU/mL (interquartile range [IQR], 95-1590) at week 48 to 566 IU/mL (IQR, 68-1495) at week 96 (p<0.001).”
- Lines 178-180 (underlined): “The median estimated glomerular filtration rate (eGFR) decreased from 94.9 mL/min/1.73m² at week 48 to 93.2 mL/min/1.73m² at week 96 (p=0.002).”
- Lines 191-196 (underlined): “The UPCR and urine $\beta 2$ -microglobulin-to-creatinine ratio continued to decline from week 48 to week 96, whereas the decrement of urine albumin-to-creatinine ratio (UACR) were similar between week 48 and week 96 (-15.3% vs. -12.5%) (Fig. 2A). The prevalence of proteinuria by urine dipstick tests decreased from 19.0% at baseline to 3.9% at week 96. Similarly, the proportion of the participants with UPCR ≥ 150 mg/g decreased from 12.1% at baseline to 7.9% at week 96 (Fig. 2B).”

- Lines 200-202 (underlined): “The median levels of triglycerides, total cholesterol, low-density lipoprotein (LDL)-cholesterol, and high-density lipoprotein (HDL)-cholesterol remained steady from week 48 to week 96.”
- Lines 211-220 (underlined): “BMD assessment was performed in 138 participants at week 96. Among these participants, the mean BMD of the spine and hip increased and peaked at week 48 (+2.0% and +1.5%, respectively), and slightly declined but still higher at week 96 than those at baseline (+0.9% and +0.5%, respectively) (Fig. 4A and Table S4). For the participants who had paired BMD assessments at baseline and week 96, we inspected the changes of T-score categories after switching to TAF-containing ART. At week 96, 44.8% of the participants who had osteopenia of the spine before switch regressed to normal T-score, and 33.3% of those who had osteoporosis of the spine regressed to osteopenia. Similar proportions of changes were observed for T-score of the hip (Fig. 4B).”
- Figure 1: We remove the data observed at week 24 and Figure 1A and 1B are merged into one figure to make it simple.
- Figure 2: After consideration, we preserve the change in quantitative proteinuria at each time point to better depict the trends (Figure 2A). However, we add a new figure (Figure 2B in the revised manuscript) to show the change in the proportion of the participants with abnormal UPCR or UACR.
- Figure 3 in the revised manuscript: The change in fasting lipids is separated into Figure 3. In this new Figure 3, we remove the data at week 24 and revise the figure to show the increment of each lipid parameter at weeks 48, 72, and 96.
- Figure 4 in the revised manuscript: We re-analyze the changes of bone mineral density by excluding the participants who did not have paired BMD assessments at baseline and week 96. In addition, the evolution of T-score category after

switching to E/c/F/TAF is analyzed. Figure 4A presents the mean percentage change in BMD. Figure 4B shows the changes of T-score categories (normal, osteopenia and osteoporosis) of the spine and hip from baseline to week 96.

Comment 2. E/C/F/TAF is rarely used nowadays in Western countries. Patients on E/C/F/TAF are usually switched to BIC/C/F/TAF (Biktyarvy) avoiding cobicistat.

Reply: Thank you for the comment. We agree with the reviewer that this study reported the virological response and metabolic effects with E/c/F/TAF. Therefore, the results on HIV suppression, change in serum creatinine and weight in this study may not be fully generalized to the patients who switch from TDF-containing regimens to TAF-containing regimens other than E/c/F/TAF. We add two sentences in Discussion to highlight this limitation (Lines 341-345, underlined): “Similarly, E/c/F/TAF is seldom used nowadays in many countries. Therefore, the results on HIV suppression, changes in serum creatinine or weight in this study may not be generalizable to people living with HIV who switch from TDF-containing ART to BIC/F/TAF or dolutegravir plus F/TAF.”

Comment 3. Could you report the weight changes after switching to E/C/F/TAF at 96 wk?. At 48wk there was a significant weight gain of 3 Kg (JAIDS 2021; 86: 473-481). This is an interesting point to report in the present manuscript.

Reply: Thank you for the query. We analyzed the weight change from week 48 to week 96 week. The median weight was the same at week 48 (72 kg, IQR 64-79 kg) and week 96 (72 kg, IQR 65-80 kg). The description of weight change is added in Results section (lines 207-209, underlined): “Of the 90 participants with follow-up weight measurements, the median weight was the same at week 48 (72 kg, IQR

64-79) and week 96 (72 kg, IQR 65-80). The median percentage increase in weight was 4.1% at week 96.”

Comment 4. Could you explain the differences between HbeAg loss and HbeAg (negative) seroconversion? They look very similar to me (Results section lines 175-176)

Reply: Thank you for the query. “HbeAg loss” refers to the change from positive HbeAg and negative anti-Hbe at baseline to negative HbeAg without development of anti-Hbe after baseline visit. “HbeAg seroconversion” refers to the change from positive HbeAg and negative anti-Hbe at baseline to negative HbeAg and positive anti-Hbe after baseline visit. To avoid confusion, we added the definitions in the Methods (Lines 373-377, underlined): “HbeAg loss” referred to the change from HbeAg positivity and anti-Hbe negativity at baseline to HbeAg negativity without the development of anti-Hbe after baseline visit. “HbeAg seroconversion” referred to the change from HbeAg positivity and anti-Hbe negativity at baseline to HbeAg negativity and anti-Hbe positivity after baseline visit.”

We also revise the description in Abstract (Line 78, underlined): “HbsAg loss occurred in 1.5% of 274 participants, and HbeAg loss or seroconversion in 14.3% of 35 HbeAg-positive participants,” and Results (Lines 173-174, underlined): “Among the 35 HbeAg-positive participants, 2 (5.7%) lost HbeAg without positive anti-Hbe antibody and 3 (8.6%) had HbeAg seroconversion.”

Comment 5. RPR titers were more elevated among anti-HDV-positive patients but not significantly (57.5% vs. 41.6%, $p=0.061$ not significant) Therefore there is a tendency but these data cannot confirm here that sexual contact is an important route of HDV transmission (Discussion section, lines 305-308)

Reply: Thank you for the comment. We add two sentences to emphasize the non-significance of our comparisons and to state that a prospective investigation to clarify the major transmission route of HDV in HBV/HIV-coinfected patients is warranted (Lines 313-316, 320-323, underlined): "Although the comparisons of RPR-positive rate among anti-HDV-positive and anti-HDV-negative participants in our study did not reach statistical significance (57.5% vs 41.6%), our findings suggested a tendency for HDV to be sexually transmitted among PLWH. Two recent studies proposed that sexual contact has become an important route for HDV transmission (35, 36). The possibility of HDV superinfection should be considered in PLWH with HBV coinfection who recently acquired sexually transmitted diseases. Moreover, high-risk patients may require regular screening for HDV. Nonetheless, prospective studies are needed to clarify the major transmission route of HDV and the association between sexual activities and acquisition of HDV in PLWH with HBV coinfection."

Comments of reviewer #2:

Comment 1. The drop outs should be discussed better in the beginning. 44 out of 274 did not complete the study but a full account is only given for 19 of them.

Reply: Thank you for the suggestion. We revise Results section to provide this information (Lines 147-154, underlined): "In this extension study, 230 (83.9%) of 274 participants completed the 96 weeks of follow-up (Fig. S1). Thirteen (4.7%) and 31 (11.3%) participants dropped out of the study before and after week 48, respectively. After week 48, 31 participants withdrew from the study because of withdrawal of consent (n=12), drug-drug interaction with E/c/F/TAF (n=8), loss to follow-up (n=8), and adverse effects (n=3, including hyperlipidemia, occurrence of non-ST-elevation

myocardial infarction, and depression). Six participants switched to other TAF-containing ART after 48 weeks and continued in the study.”

Comment 2. Discuss long term impact of increased levels of fasting lipids for liver pathology in HBV infected people.

Reply: Thank you for the comment. We agree with the reviewer that impacts of increased fasting lipids on liver pathology is an important issue and should be discussed for PLWH with HBV coinfection. We add several sentences in the fourth paragraph of Discussion (lines 302 to 309, underlined) to discuss this point, as follows: “Another concern is the impact of increased fasting lipids on the liver. Hyperlipidemia was found to be associated with fatty liver disease in PLWH with HBV coinfection. Steatosis alone was independently correlated with higher ALT after adjustment for HBV DNA level. Prior studies revealed that use of TAF or integrase inhibitors but not TDF increased the risk of steatosis in PLWH. However, the changes in fasting lipids and liver pathology after modifying ART regimens and their long-term clinical impact have not been investigated in PLWH with HBV coinfection.” Three reference papers were cited here: **Khalili et al. Clin Infect Dis. 2021 73(9):e3275-e3285, Bischoff et al. EClinicalMedicine. 2021 40:101116., Riebensahm et al. Open Forum Infect Dis. 2022 9(11):ofac538.**

Comment 3. Shortcomings are reported but dampen impact of the study (eg. No monitoring of lipid lowering drugs)

Reply: We agree with the reviewer that the lack of documentation of lipid-lowering drugs precluded us from clarifying the contributing factor of lipid changes. This significant limitation is described in Discussion section. At present, studies comparing the proportion of initiating lipid-lowering agents among PLWH starting TAF- or

TDF-containing ART showed conflicting results. A more extensive surveillance is warranted to understand the prescription of lipid-lowering drugs among PLWH with different underlying diseases, ART regimens, baseline LDL levels, and risks of cardiovascular disease. We add a sentence in Discussion to strengthen this point (line 300 to 302, underlined): “While the conflicting results may be related to different study designs and populations, assessment of lipids and the risk for cardiovascular disease is warranted for timely initiation of lipid-lowering medications, particularly when PLWH who have achieved viral suppression are ageing. The prescription pattern of lipid-lowering drugs among PLWH in real-world setting should also be monitored and evaluated for the adherence to the guidelines on primary prevention of cardiovascular disease.”

Comment 4. Typo’s: Line 84: having instead of had; Line 319: were not included

Reply: Thank you for the correction. We correct the grammatical errors in Line 93 and Line 331 per review’s suggestion.

Comments of reviewer #3:

Reviewer #3: The authors studied the efficacy and safety of switching from tenofovir disoproxil fumarate-based antiretroviral therapy to coformulated elvitegravir, cobicistat, emtricitabine, and tenofovir alafenamide (E/C/F/TAF) in HIV/hepatitis B virus (HBV)-coinfected Asian patients from Baseline to Week 96.

GENERAL OBSERVATIONS

Comment 1. Similar study, as also mentioned in Ref 12, has already been conducted and published. In terms of results, you observed in this new unpublished paper same virologic, serologic and biochemical responses compared to the published study; although, in this new paper, you kept observing patients up to 96 weeks. Considering

these results, this new paper should be re-adapted in a form that includes a brief summary of the previous study (by reporting the observed results at 24 and 48 weeks), and the new results related with follow up at 72 and 96 weeks. Repeating all the work already done in the published paper (from baseline to 48 hours) is unnecessary and can be confusing (e.g. LINE 144, LINE 341).

In conclusion, it is recommended to edit this manuscript by making a summary of the results obtained so far (baseline and week 48 from the study of REF. 12) and to describe the results observed in week 72 and 96, which are the unpublished results.

Reply: Thank you for your comment. We revised the Results section per the reviewer's suggestion. The observed data from baseline to 48 weeks are briefly summarized in the first paragraph, and this study focus on describing data observed between weeks 48 and 96. We also revise or add Figures 1 to 4. The data from baseline to week 48 are removed in some of the figures, and we present the changes in proteinuria and bone mineral density between weeks 48 and 96 with different approaches. The revised sentences and figures are listed, as follows (underlined):

- Lines 140-146, summary of previously published study (underlined): “A total of 274 HIV/HBV-coinfected participants were included. The clinical characteristics of the participants and the study outcomes at week 48 have been previously presented (12). In summary, HIV-suppressed PLWH with HBV coinfection who switched from TDF-containing ART to E/c/F/TAF achieved high rates of HBV DNA <20 IU/mL (89.8%) and HIV RNA <50 copies/mL (94.2%) at week 48. Also, the participants showed improvement of proteinuria and BMD of the spine and hip but increased fasting lipid levels at weeks 24 and 48.”
- Lines 155-157 (underlined): “The proportions of the participants with HBV DNA \geq 20 IU/mL at weeks 72 and 96 were 5.8% and 5.1%, respectively.”

- Lines 161-163 (underlined): “One participant who had high HBV DNA load (11400 IU/mL) at week 72 also showed an increase of plasma HIV RNA from <50 to 1231078 copies/mL, suggesting poor treatment adherence”
- Lines 168-169 (underlined): “The median HBsAg level continued to decrease from 656 IU/mL (interquartile range [IQR], 95-1590) at week 48 to 566 IU/mL (IQR, 68-1495) at week 96 (p<0.001).”
- Lines 178-180 (underlined): “The median estimated glomerular filtration rate (eGFR) decreased from 94.9 mL/min/1.73m² at week 48 to 93.2 mL/min/1.73m² at week 96 (p=0.002).”
- Lines 191-196 (underlined): “The UPCR and urine β 2-microglobulin-to-creatinine ratio continued to decline from week 48 to week 96, whereas the decrement of urine albumin-to-creatinine ratio (UACR) were similar between week 48 and week 96 (-15.3% vs. -12.5%) (Fig. 2A). The prevalence of proteinuria by urine dipstick tests decreased from 19.0% at baseline to 3.9% at week 96. Similarly, the proportion of the participants with UPCR \geq 150 mg/g decreased from 12.1% at baseline to 7.9% at week 96 (Fig. 2B).”
- Lines 200-202 (underlined): “The median levels of triglycerides, total cholesterol, low-density lipoprotein (LDL)-cholesterol, and high-density lipoprotein (HDL)-cholesterol remained steady from week 48 to week 96.”
- Lines 210-219 (underlined): “BMD assessment was performed in 138 participants at week 96. Among these participants, the mean BMD of the spine and hip increased and peaked at week 48 (+2.0% and +1.5%, respectively), and slightly declined but still higher at week 96 than those at baseline (+0.9% and +0.5%, respectively) (Fig. 4A and Table S4). For the participants who had paired BMD assessments at baseline and week 96, we inspected the changes of T-score categories after switching to TAF-containing ART. At week 96, 44.8% of the

participants who had osteopenia of the spine before switch regressed to normal T-score, and 33.3% of those who had osteoporosis of the spine regressed to osteopenia. Similar proportions of changes were observed for T-score of the hip (Fig. 4B)."

- Figure 1: We remove the data observed at week 24 and Figure 1A and 1B are merged into one figure to make it simple.
- Figure 2: After consideration, we preserve the change in quantitative proteinuria at each time point to better depict the trends (Figure 2A). However, we add a new figure (Figure 2B in the revised manuscript) to show the change in the proportion of the participants with abnormal UPCR or UACR.
- Figure 3 in the revised manuscript: The change in fasting lipids is separated into Figure 3. In this new Figure 3, we remove the data at week 24 and revise the figure to show the increment of each lipid parameter at weeks 48, 72, and 96.
- Figure 4 in the revised manuscript: We re-analyze the changes of bone mineral density by excluding the participants who did not have paired BMD assessments at baseline and week 96. In addition, the evolution of T-score category after switching to E/c/F/TAF is analyzed. Figure 4A presents the mean percentage change in BMD. Figure 4B shows the changes of T-score categories (normal, osteopenia and osteoporosis) of the spine and hip from baseline to week 96.

Comment 2. Please check the Reference 6 in LINE 113 and make sure it is related with the Ref. 6 of the REFERENCE section

Reply: We correct the format of reference 6 in the main text (Line 111). The reference 6 is "Guidelines for the Use of Antiretroviral Agents in Adults and Adolescents with HIV," published by the United States Department of Health and Human Services.

Comment 3. LINE 132-133: Reference required; please add one or more references related with the low evaluation of the virologic and serologic parameters in the co-infected patients.

Reply: Thank you for the comment. We review the studies about virologic and serologic responses of HBV to TAF-containing ART in HIV/HBV-coinfected patients (Sarowar et al. *J Acquir Immune Defic Syndr.* 2022;91(4):368-372, Surial et al. *J Acquir Immune Defic Syndr.* 2020;85(2):227-232., Gallant et al. *J Acquir Immune Defic Syndr.* 2016;73(3):294-298.). As shown in the table below, the reported parameters by each study were incomplete, and most studies did not analyze the quantitative change in HBsAg. We revise the sentence in Introduction to more specifically describe this data gap. The above-mentioned references are cited (Ref. 16-18 in the revised manuscript) (Lines 128-131, underlined): “However, the evaluation of serologic end points of HBV to TAF-containing ART in PLWH with HBV coinfection were often incomplete, especially for the quantitative HBsAg, and long-term observation has been lacking (16-18).”

Author (Year)	Patient number	Undetectable HBV DNA (%)	Change in quantitative HBsAg	HBV serologic markers
Sarowar et al. (2022)	82	98%	Not available	HBsAg loss: Not available HBeAg seroconversion: 3%
Surial et al. (2020)	106	97%	Not available	HBsAg loss: 8.7% HBeAg seroconversion: Not available
Gallant et al.	72	91.7%	Not available	HBsAg loss: 2.9 %

(2016)				HBeAg loss: 3.3% HBeAg seroconversion: 3.3%
--------	--	--	--	--

February 28, 2023

Dr. Chien-Ching Hung
National Taiwan University Hospital
Department of Internal Medicine
7, Chung-Shan South Road
Taipei 100
Taiwan

Re: Spectrum05125-22R1 (Week 96 results of switching from tenofovir disoproxil fumarate-based antiretroviral therapy to coformulated elvitegravir, cobicistat, emtricitabine and tenofovir alafenamide among HIV/HBV-coinfected patients)

Dear Dr. Chien-Ching Hung:

Your manuscript has been accepted, and I am forwarding it to the ASM Journals Department for publication. You will be notified when your proofs are ready to be viewed.

Sincerely,

Yuan Pin Hung
Editor, Microbiology Spectrum
